# Syphilitic Cholangiopathy Mimicking Primary Sclerosing Cholangitis

**DOI:** 10.3390/idr17020023

**Published:** 2025-03-06

**Authors:** Adriana Gregušová, Michal Gergel, Miroslav Žigrai

**Affiliations:** 11st Department of Internal Medicine, Slovak Medical University, Bratislava University Hospital, 83303 Bratislava, Slovakia; adriana.gregusova@kr.unb.sk (A.G.); miroslav.zigrai@kr.unb.sk (M.Ž.); 21st Department of Surgery, Slovak Medical University, Bratislava University Hospital, 83303 Bratislava, Slovakia

**Keywords:** syphilis, hepatitis, sclerosing cholangitis

## Abstract

Introduction: Syphilis is a sexually transmitted disease with variable symptoms, often imitating various other disorders. Syphilis progresses through primary, secondary, latent, and tertiary stages, each with distinct clinical manifestations. A sudden rise in serum hepatic enzyme levels and imaging findings that mimic sclerosing cholangitis, both associated with a positive response to targeted antibiotic treatment, may indicate a diagnosis of acute syphilitic hepatitis. Case Presentation: We report a case of early syphilis in the secondary stage, manifesting as sclerosing-cholangitis-like changes shown on ultrasonography, MR, and CT. Narrow-spectrum antibiotic therapy with procaine benzylpenicillin led to a consistent decrease in and normalization of levels of serum bilirubin and other markers of hepatic injury. Repeated sonography and MR cholangiography showed minimal residual changes in the intrahepatic biliary tree. Conclusions: Infection with *Treponema pallidum* is one of the rare causes of secondary cholangitis. As the incidence of syphilis is rising worldwide, it should be considered as a differential diagnosis, especially for patients with high-risk sexual behavior and for whom there are laboratory findings of cholestatic or mixed cytolytic and cholestatic hepatitis, particularly if associated with exanthema, pharyngitis, and lymphadenopathy.

## 1. Introduction

A wide spectrum of cholangiopathies is known, including, apart from primary sclerosing cholangitis (PSC) and primary biliary cholangitis (PBC), secondary sclerosing cholangitis (SSC) with an identifiable etiology. SSC includes autoimmune IgG4-associated cholangitis, ischemic cholangitis, eosinophilic cholangitis, recurrent pyogenic cholangitis, virus-associated cholangitis (AIDS and COVID-19), parasitic cholangiopathy (*Cryptosporidium*, *Microsporidium*, and *Ascaris lumbricoides*), congenital syndromes (Caroli disease), portal biliopathy, and toxin-induced bile duct injury. A correct diagnosis of PSC is made based on the exclusion of secondary causes of cholangiopathy [1].

An important point is that, in contrast to PSC, wherein liver transplantation is an important therapeutic option, SSC can be successfully treated by effectively managing its causative origin. Accurate classification of sclerosing cholangitis, therefore, requires precise differential diagnosis, including a comprehensive spectrum of imaging methods [1,2,3,4]. Early diagnosis of infectious SSC is extremely important in order to provide directed and effective anti-infective treatment.

The development of a liver injury is common during the secondary phase of syphilis; however, syphilis-induced secondary cholangitis has been described as a very rare presentation of this disease. Because of this rareness, it is not included in common guidelines for the differential diagnosis of SSC [1]. The anticipated pathomechanism appears to be associated with cholestatic liver injury caused by pericholangial inflammation and direct biliary tract infiltration through portal circulation following *Treponema pallidum* transmission during anal intercourse [5].

## 2. Case Presentation

A 35-year-old male with a BMI of 25.2 with no known personal or family history of relevant diseases and engaged in a same-sex relationship was admitted to the hepatology unit due to several days of jaundice, dark urine, and bright stool. He also complained of general weakness, anorexia, a sore throat, night sweats, and weight loss (approximately 8 kg per month) for 6 weeks prior to the onset of jaundice. He reported an erythematous rash on both his upper and lower limbs. Physical examination on admission revealed punctate, painless, and non-itchy exanthema on the forearms, hands, palms, and lower limbs, particularly on both soles. Laboratory investigations showed total hyperbilirubinemia amounting to 259 nmol/L (reference range: 0–21 nmol/L), conjugated hyperbilirubinemia amounting to 194 nmol/L (reference range: 0–3.4 nmol/L), and increased serum activity of alkaline phosphatase at 17 µkat/L (ALP, reference range: 0.71–1.92 µkat/L), gamma-glutamyl transferase at 6.7 µkat/L (GGT, reference range: 0–0.92 µkat/L), aspartate aminotransferase at 3.6 µkat/L (AST, reference range: 0–0.85 µkat/L), and alanine aminotransferase at 5.56 µkat/L (ALT, reference range: 0–0.85 µkat/L).

Ultrasonography showed hepatomegaly and intrahepatic biliary duct dilation with focal, irregular wall thickening. There were no signs of extrahepatic biliary obstruction. MR cholangiography clarified several strictures of the intrahepatic ducts with apparent post-stenotic dilatations in both liver lobes, more significant on the right side. Dilatation of the extrahepatic ducts was again not present (Figure 1).

Viral and autoimmune etiologies were investigated and turned out to be negative: HAV (hepatitis A virus), HBV (hepatitis B virus), HCV (hepatitis C virus), CMV (cytomegalovirus), EBV (Ebstein–Barr virus), and HIV (human immunodeficiency virus) were serology-negative; AMAs (anti-mitochondrial antibodies), ANAs (anti-nuclear antibodies), pANCAs (perinuclear anti-neutrophil cytoplasmic antibodies), ALKM (anti-liver kidney microsomal antibody), AEmAs (anti-endomysium antibodies), ALMAs(anti-liver cell membrane antibodies), and ANLAs (anti-nucleolar antibodies) were all IgM-negative (ANLA results were IgG-positive).

The patient denied abusing alcohol or illegal drugs, and there were no objective signs of such behavior. He was not taking prescribed medication either. He was taking an over-the-counter plant-based medicine containing alpha-pinene, beta-pinene, cineole, menthone, menthol, and borneol. Since he took the first dose 4 days prior to hospitalization, after the onset of symptoms, drug-induced liver injury was evaluated as improbable.

According to these negative results, following current guidelines [1], we determined there was no evidence of a secondary etiology of the sclerosing cholangitis. Based on the sex and age of the patient, US and MRI findings, and the lack of evidence of a secondary etiology, primary sclerosing cholangitis was determined to be the most probable diagnosis. As the patient’s bilirubin levels were continuously rising (Figure 2), initial examination steps toward liver transplantation were initiated. Esophagogastroscopy revealed mild gastroesophageal reflux disease, while colonoscopy showed negative results.

At this stage, initial pretransplant results were obtained, surprisingly showing positive serological results for syphilis. Both a rapid plasma reagin test (RPR) (1:32 titer) and the *Treponema pallidum* particle assay test (TPPA) (1:38) turned out positive and confirmed the diagnosis of syphilis. As *Treponema pallidum* is known to be a causative agent of secondary sclerosing cholangitis, further pretransplant examinations were discontinued, and empirical causative therapy was initiated with procaine benzylpenicillin at a daily dose of 1.5 million units for 22 days. As a result, serum bilirubin levels and the levels of other hepatic markers decreased significantly and reached normal levels (Figure 2). A follow-up MRCP after 3 months showed only minor residual changes in the intrahepatic ducts of the left lobe, suggesting an almost complete cure (Figure 3 and Figure 4).

## 3. Discussion

Syphilis is a sexually transmitted disease caused by the intracellular Gram-negative spirochete *Treponema pallidum* subsp. *pallidum*. Its initial stage, primary syphilis, presents either as a solitary painless ulcer or multiple lesions typically located on the genitals. These lesions may resolve spontaneously and go unnoticed by the patient. In the next phase, secondary syphilis, dissemination of the pathogen occurs, causing general weakness, anorexia, weight loss, skin lesions (typically a non-itchy rash), and damage to multiple organs. If untreated, the disease evolves after several weeks to the latent stage, which can progress to the third stage, tertiary syphilis, after several years. Late tertiary syphilis can manifest years after the infection, affecting the bones, skin, gingiva, cardiovascular system, and/or central nervous system by forming specific gummas [1,5,6].

*Treponema pallidum*-associated liver injury is reported to occur in about 50% of cases, predominantly in the secondary stage. Syphilitic liver injury is often presented solely with elevated hepatic markers, typically cholestatic enzymes (GGT and ALP), while serum bilirubin levels remain normal or are only slightly increased [2]. Cytolytic hepatitis occurs very rarely [7]. An icteric course of syphilitic hepatitis, which occurred in our case, has been observed less frequently, in about 30% of cases. Fulminant liver failure is extremely rare, with only two known cases treated via liver transplantation, both resulting in fatal outcomes [8,9]. Histology typically shows necrosis of hepatocytes around the central vein, mild non-specific granulomatous changes, and pericholangitis with lymphocyte infiltrates, which explains the cholestatic character of syphilitic hepatitis [2,9].

The diagnosis of syphilis can be either direct, proving the presence of the spirochetes themselves, or indirect, based on serologic antibody detection. The classic morphological test (darkfield examination, DFE) detects the specific spiral shape of *Treponema pallidum* and may be useful for diagnosing early primary lesions. However, the ability tosample these lesions is sometimes limited due to their rapid dissolution, and sensitivity varies between 71% and 100% [10]. Therefore, PCR-based molecular diagnosis is currently the preferred direct method [11]. An indirect diagnosis is made utilizing nontreponemal (NTT) and treponemal (TT) tests, depending on the targeted antigen [12]. Non-treponemal tests use cardiolipin (or lecithin and cholesterol) as antigens, while treponemal tests detect the presence of antibodies against specific treponemal antigens. Non-treponemal tests can detect antibodies in blood and cerebrospinal fluid 2–3 weeks and4–8 weeks after infection, respectively. Anti-lipoid (anti-cardiolipin) antibodies, which are induced by syphilitic tissue damage, target tissue antigens, making them prone to false-positivity due to other infections or collagenoses. Therefore, it is necessary to perform specific treponemal tests simultaneously. Specific anti-treponemal antibodies form 5–6 weeks after infection, and their concentration significantly rises during the secondary stage of the disease. For routine treponemal antigen detection, either agglutination methods (TPPA and TPHA) or immunoassay methods (ELISA, CLIA, ECLIA, and CMIA) are used. False-positives can sometimes occur due to cross-reactivity with other *Treponema* species or other underlying disorders (e.g., autoimmune conditions).

The sensitivity of NTT is 62–78% for primary syphilis, 97–100% for secondary syphilis, and 82–100% for early latent syphilis. The sensitivity for tertiary syphilis is insufficient, at 47–64% [13,14].

A liver biopsy is not considered necessary for the definitive diagnosis of syphilitic hepatitis as long as there is a positive response to specific therapy. This is due to the histological changes’ nonspecific nature, which makes it difficult to differentiate syphilitic hepatitis from drug-induced liver injury. Histological signs of syphilitic hepatitis may include lymphoplasmacytic and granulomatous portal inflammation with bile duct damage [15,16]. Direct identification of spirochetes in liver tissue is challenging and rarely successful, whether through immunohistochemical methods or Warthin–Starry staining [17,18]. Ultrasound and CT scanning usually show only nonspecific pathologies, such as hepatosplenomegaly, hyperechoic liver, and intraabdominal lymphadenopathy [19,20]. Imaging changes to the bile ducts have been reported infrequently. According to available sources, magnetic resonance cholangiopancreatography (MRCP) has shown only normal findings in the biliary tree and liver in cases of syphilitic hepatitis, with no reports of bile duct dilation. Hussain et al. did not detect biliary tree dilation on MRCP scans in a patient with syphilitic hepatitis [20]. There are limited data available on changes in the large bile ducts in syphilitic hepatitis. Published cases of icteric syphilitic hepatitis have reported PSC-like changes on MRCP scans in patients with syphilitic hepatitis; however, these patients were also infected with HIV, which is a known cause of secondary sclerosing cholangitis [17,20,21]. In older reports, MRCP was either not performed or the results were not published. In all published cases, causative anti treponemal therapy led to substantial or complete resolution of cholangitis.

No general consensus on the diagnostic criteria for syphilitic hepatitis has been accepted thus far. Mullick et al. (2004) suggested that positivity of four criteria is necessary to establish a diagnosis of syphilitic hepatitis: abnormal serum levels of liver enzymes (predominantly cholestatic); serologic proof of syphilis infection; exclusion of other etiologies of liver damage; and normalization of liver enzyme levels after adequate antibiotic treatment [22] (Table 1). *Treponema pallidum* infection was not considered for our patient during the early differential diagnostic process, although the observed skin changes could have been viewed as a suspicious sign. This oversight was due to the low prevalence of syphilis in our region, as well as the lack of information regarding the patient’s sexual behavior, which he did not disclose during the initial interview. Given the patient’s age, which is typical for primary sclerosing cholangitis (PSC) manifestation, along with concordant MRCP findings, PSC was considered the most probable diagnosis. The positive serological results for syphilis constituted an unexpected finding during the initial pre-transplant screening. This discovery led to prompt antibiotic treatment, followed by a gradual decrease in serum bilirubin and hepatic marker levels as well as significant, almost complete restitution of bile duct pathology.

## 4. Conclusions

Various infections are known to cause secondary sclerosing cholangitis, including *Treponema pallidum* infection, which is typically associated with secondary-stage syphilis. This should be taken into consideration due to the rising prevalence of syphilis worldwide. The presence of laboratory signs of cholestatic or mixed cytolytic–cholestatic hepatitis, particularly when accompanied by typical exanthema, pharyngitis, and lymphadenopathy, especially in patients with high-risk lifestyles and sexual behaviors, should raise suspicion and trigger a differential diagnosis of syphilitic infection.

## Figures and Tables

**Figure 1 idr-17-00023-f001:**
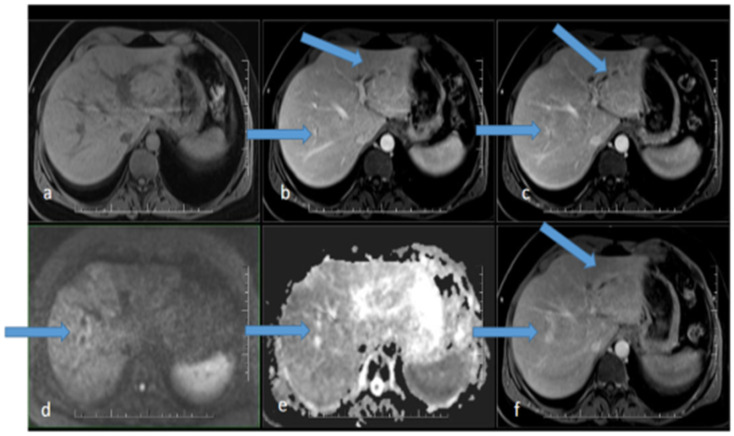
MR scan of the liver before the initiation of antibiotic therapy: (**a**) native image; (**b**) arterial phase; (**c**) venous phase; (**d**) diffusion-weighted imaging; (**e**) ADC map; (**f**) delayed phase. Apparent focal dilatation of the intrahepatic bile ducts is marked with arrows in the left and right lobes in the arterial (**b**), venous (**c**), and delayed phases, showing gradual opacification of the duct walls and peribiliary spaces, respectively. Increased signal in the duct wall area is indicated in diffusion-weighted imaging (**d**), with an iso-signal in the ADC map (**e**).

**Figure 2 idr-17-00023-f002:**
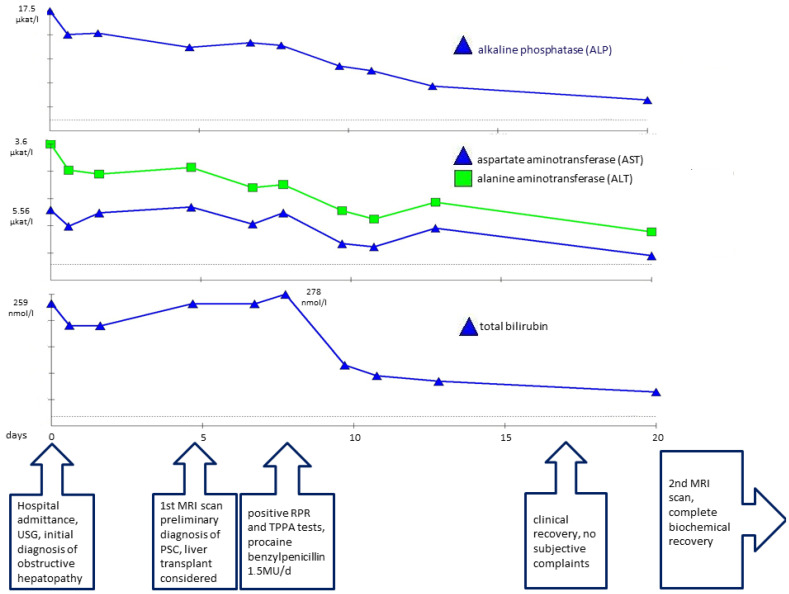
Plasmatic concentrations of alkaline phosphatase (ALP), aspartate aminotransferase (AST), alanine aminotransferase (ALT), and total bilirubin associated with timeline of disease progression.

**Figure 3 idr-17-00023-f003:**
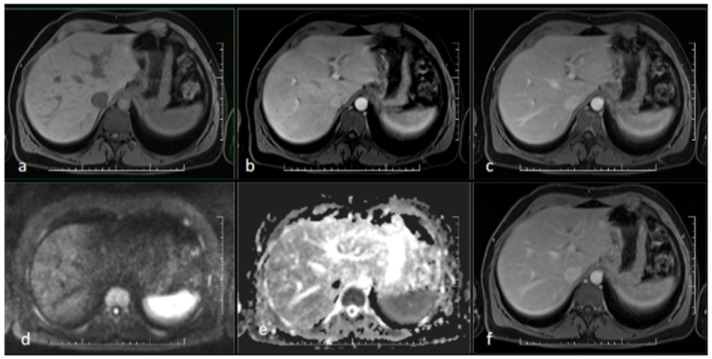
MR scan of the liver after antibiotic treatment: (**a**) native image; (**b**) arterial phase; (**c**) venous phase; (**d**) diffusion-weighted imaging; (**e**) ADC map; (**f**) delayed phase. Narrowing of the originally markedly dilated intrahepatic biliary ducts and regression of pathological opacification are noticeable in the arterial (**b**), venous (**c**), and delayed (**f**) phases. Regression of the pathologic signal was detected in diffusion-weighted imaging (**d**).

**Figure 4 idr-17-00023-f004:**
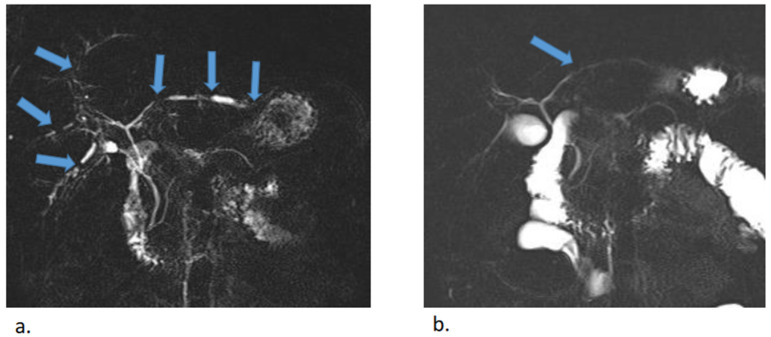
MR cholangiopancreatography before (**a**) and after (**b**) antibiotic treatment. Multiple uneven stenoses of the intrahepatic biliary ducts were detectable in both lobes of the liver (blue arrows). Almost complete regression of stenoses was observed after treatment, with minor residual stenoses in the left lobe.

**Table 1 idr-17-00023-t001:** Mullick’s diagnostic criteria for syphilitic hepatitis [22].

Mullick’s Diagnostic Criteria for Syphilitic Hepatitis:
Abnormal liver enzyme levels; Serological evidence for syphilis; Exclusion of other causes of liver disease; Liver enzyme levels returning to normal after appropriate antimicrobial therapy is performed.

## Data Availability

All data is included in patient’s records and cannot be published due to personal data protection policy.

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
