# Peer review of "Syphilitic Cholangiopathy Mimicking Primary Sclerosing Cholangitis"

_2036-7449, 2025, doi:10.3390/idr17020023_

Round 1
Reviewer 1 Report
Comments and Suggestions for Authors
The manuscript presents an interesting and well-documented case of syphilitic cholangiopathy, an exceptionally rare manifestation of secondary syphilis, misdiagnosed initially as primary sclerosing cholangitis (PSC). The case is well-structured, with clear clinical reasoning, imaging findings, laboratory data, and a successful treatment outcome.
The study highlights an important differential diagnosis in hepatobiliary disorders, particularly given the increasing incidence of syphilis worldwide. The discussion effectively places the findings in context with existing literature, though some areas could benefit from additional depth, clarity, and improved conciseness.
Abstract:
Consider the following rephrasing:
"Syphilis progresses through primary, secondary, latent, and tertiary stages, each with distinct clinical manifestations."
Introduction:
The introduction should briefly summarize the clinical importance of syphilitic hepatitis and its potential to mimic PSC, setting the stage for the case report.
Current introduction is too broad on cholangiopathies; focusing on why syphilis can present as cholangiopathy would improve clarity.
Case presentation
Condense laboratory values into a table for better readability.
Highlight critical decision points leading to misdiagnosis and eventual correct diagnosis.
The pre-transplant screening discovery of syphilis should be emphasized more explicitly.
Discussion
The diagnostic algorithm for syphilitic hepatitis (Mullick’s criteria) should be explicitly summarized in a table or figure for clarity.
Figures:
A timeline of disease progression (symptom onset, misdiagnosis, correct diagnosis, treatment, follow-up) would enhance clarity.
When discussing viral-associated cholangiopathies, you can reference these studies as examples of how hepatitis C affects metabolic and biochemical parameters: doi: 10.47162/RJME.61.4.20 and 10.31925/farmacia.2022.4.9.
Author Response
Thank you very much for your review, all suggestions have been appreciated.
Comment: Consider the following rephrasing: "Syphilis progresses through primary, secondary, latent, and tertiary stages, each with distinct clinical manifestations."
Response: Suggested rephrasing makes the abstract more readable, sentences have been replaced.
Comment: The introduction should briefly summarize the clinical importance of syphilitic hepatitis and its potential to mimic PSC, setting the stage for the case report.
Response: Information on syphillitic cholangiopathy and its importance has been added, along with suitable citation
Comment: Current introduction is too broad on cholangiopathies; focusing on why syphilis can present as cholangiopathy would improve clarity.
Response: The report is conceived as hepatologic point of view, therefore the broad introduction. Nevertheless, modification from the previous point might make this point more clear as well.
Comment: Condense laboratory values into a table for better readability.
Response: Important laboratory values have been added to Figure 4
Comment: Highlight critical decision points leading to misdiagnosis and eventual correct diagnosis.
Response: Differential diagnosis process has been rephrased, emphasizing the reason of original misdiagnosis.
Comment: The pre-transplant screening discovery of syphilis should be emphasized more explicitly.
Response: Syphillis diagnosis has been described more into details to clarify the process
Comment: The diagnostic algorithm for syphilitic hepatitis (Mullick’s criteria) should be explicitly summarized in a table or figure for clarity.
Response: Table has been added
Comment: A timeline of disease progression (symptom onset, misdiagnosis, correct diagnosis, treatment, follow-up) would enhance clarity.
Response: Timeline has been added and connected to curves of basic laboratory values in Figure 4
Comment: When discussing viral-associated cholangiopathies, you can reference these studies...
Response: Thank you for your suggestion. Differential diagnosis of hepatitis C clearly is an issue when diagnosing patients with syphillis and liver injury, however it was ruled out quite early during our diagnostic process. Anyway, we will keep your suggestion in mind for our next paper in preparation.
Reviewer 2 Report
Comments and Suggestions for Authors
Introduction line 37 - reference is misisng
Line 44- add pallidum and italicize
Case report- please report if patient has any family history of autoimmune liver disease, was he on any medication to prevent HIV transmission (Pre exposure prophylaxis), what was his BMI and if the patient used any illicit drugs/alcohol etc?
Please report what tests were exactly done for syphilis testing on your patients and what were the exact results?
Line 67, 68- abbreviations must be explained
Line 79- italicize treponema pallidum
Figure 4- it would be useful to add values of AST/ALT and INR
Line 106- italicize pathogen name
Discussion should compare this case with the ones previously reported in the literature
Comments on the Quality of English Language
needs some corrections in grammar and syntax
Author Response
Thank you very much for your review, all suggestions have been appreciated.
Comment: Introduction line 37 - reference is misisng
Response: adequate citation has been added
Comment: Line 44- add pallidum and italicize
Response: Missed this one, corrected
Comment: Case report- please report if patient has any family history of autoimmune liver disease, was he on any medication to prevent HIV transmission (Pre exposure prophylaxis), what was his BMI and if the patient used any illicit drugs/alcohol etc?
Response: information on family history and basic anthropometry have been added. There was no evidence of HIV infection in our patient, neither in history nor in current disease, pre exposure prophylaxis in risky behaviour people is not a common practice in our coutry due to generally low prevalence of HIV infection. There was no evidence of drug or alcohol abuse, this information has been added. It is surely important. thank you for your notice.
Comment: Please report what tests were exactly done for syphilis testing on your patients and what were the exact results?
Response: Data has been added
Comment: Line 67, 68- abbreviations must be explained
Response: Abbreviation list has been completed and explanations have been added to the text
Comment: Line 79- italicize treponema pallidum
Response: corrected
Comment: Figure 4- it would be useful to add values of AST/ALT and INR
Response: AST/ALT levels have been added, INR values were within normal margins all the time, ALP has been added instead as a better marker of obstructive hepatopathy
Comment: Line 106- italicize pathogen name
Response: corrected
Comment: Discussion should compare this case with the ones previously reported in the literature
Response: Most reports show secondary cholangitis in patients with both HIV and syphillis infection, in contrast to our case. Among those reporting on solely syphillitic cholangitis there is no MRI diagnostics provided. These points are involved in the discussion. Some information on therapy and outcomes has been added.
Round 2
Reviewer 2 Report
Comments and Suggestions for Authors
I would like to thank the authors for their detailed revisions
Comments on the Quality of English LanguageCan be improved, there are several grammatical errors